# REASONING DIFFUSION FOR UNPAIRED TEXT-IMAGE TO VIDEO GENERATION

## ABSTRACT

Text-image to video generation aims to synthesize a video conditioned on the given text-image inputs. Nevertheless, existing methods generally assume that the semantic information carried in the input text and image tends to be perfectly paired and temporally aligned, occurring simultaneously in the generated video. As such, existing literature struggles with "unpaired" text-image inputs in the more universal and realistic scenario where i) the semantic information carried by the text and image may occur at different timestamps and ii) the condition image can appear at an arbitrary position rather than the first frame of the synthesized video. Video generation under this unpaired setting poses an urgent need to conduct reasoning over the intrinsic connections between the given textual description and referred image, which is challenging and remains unexplored. To address the challenge, in this paper we study the problem of unpaired text-image to video generation for the first time, proposing ReasonDiff, a novel model for accurate video generation from unpaired text-image inputs. Specifically, Reason-Diff designs a VisionNarrator module to harness the powerful reasoning abilities of a multi-modal large language model to analyze the conditioned unpaired text-image inputs, producing coherent per-frame narratives that temporally align them. Building upon this VisionNarrator module, ReasonDiff further introduces a novel AlignFormer module, which employs a Multi-stage Temporal Anchor Attention mechanism to predict frame-wise latent representations. These reasoning-enhanced latents are subsequently fused with the condition frame, providing structured guidance throughout the video generation process. Extensive experiments and ablation studies demonstrate that ReasonDiff significantly beats state-of-the-art baselines in terms of video generation quality with unpaired text-image inputs. For ease of illustration, the generated video samples can be found at the following address: `https://reasondiff.github.io/`.

## 1 INTRODUCTION

Video generative models (Ho et al., 2022; Blattmann et al., 2023b;a; Zhang et al., 2025; Guo et al., 2023; HaCohen et al., 2024) have emerged as powerful tools to produce high quality videos by iteratively refining noise through a stochastic process. By leveraging powerful backbone architectures such as Diffusion Transformer (DiT) (Peebles & Xie, 2023) or U-Net (Ronneberger et al., 2015), these models excel at capturing complex dynamics, enabling a wide range of generative tasks. Among these tasks, text-image to video generation (Wan et al., 2025; Yang et al., 2024; Hu et al., 2022; Ni et al., 2023) has become particularly important, as it targets at synthesizing videos that faithfully reflect both the given image and textual inputs.

However, most existing models heavily rely on the assumption that the semantic information carried in the conditioned input image and text are perfectly paired and temporally aligned, following a paradigm where both modalities describe the event with the same semantic meaning and the generated video is expected to begin with the input image (serving as the first frame). This strong reliance on paired text-image inputs limits the flexibility of these models, being impractical in real-world scenarios where such alignment is often absent because the user-provided conditions may not be inherently paired. For instance, the "unpaired" scenario may occur when there exist time differences between the events described by the image and text, which leads to the seeming unrelatedness of the semantic information carried in the two modalities. This may result in a failure to reason about

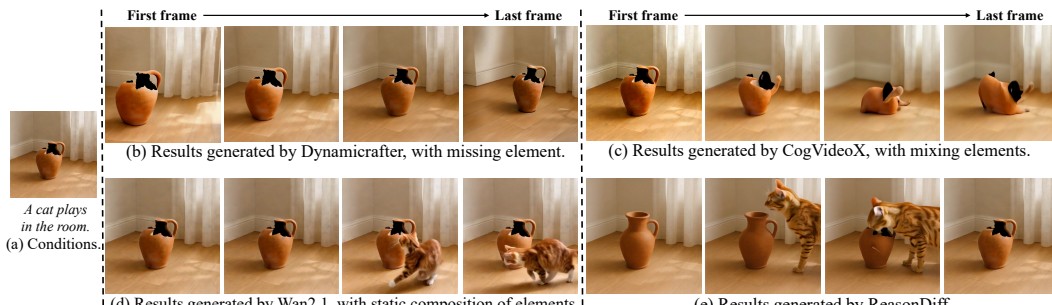

First frame ⟶ Last frame    First frame ⟶ Last frame

(b) Results generated by Dynamicrafter, with missing element.   (c) Results generated by CogVideoX, with mixing elements.

*A cat plays in the room.*
(a) Conditions.

(d) Results generated by Wan2.1, with static composition of elements.   (e) Results generated by ReasonDiff.

Figure 1: Comparison of generated results from different models with unpaired text-image inputs: i) the textual prompt *A cat plays in the room* and ii) the visual condition image *a broken vase* in Figure 1(a). Intermediate frames are selected for the convenience of presentation. Our proposed ReasonDiff has the best generation result with a visually and semantically coherent video.

their temporal connections and to bridge the semantic gap between the two modalities. As such, existing approaches will struggle to generate a coherent video both visually and semantically when encountering unpaired text-image inputs. Consider the example illustrated in Figure 1, where the model is expected to generate a video based on the unpaired inputs, *i.e.*, input text prompt as *A cat plays in the room* and input condition image as *a broken vase*. The semantic meaning carried in these two inputs may seem unrelated, but they imply an underlying connection that *the vase is broken by the cat*. When given these conditions, the model is required to recover the whole scene, and the most plausible position for the condition image is somewhere near the end of the scene. Figure 1(b) shows that existing methods tend to be dominated by one of the conditions, most frequently the image, and losing critical elements described in the text prompt. Figures 1(c) and 1(d) further highlight failure cases in which the model fails to establish meaningful relationships between the two conditions. In such cases, the outputs either represent a superficial blending of modalities or a mere juxtaposition of elements, ultimately producing videos that lack semantic coherence and visual clarity. In contrast, the video generated by ReasonDiff, shown in Figure 1(e), faithfully adheres to the given conditions: the vase remains intact initially and only breaks after interacting with the cat. This problem is challenging, as it requires inferring a plausible scene from unpaired text-image inputs, while also integrating the high-level reasoning information into the generation process.

To tackle the above challenges, in this paper we propose a novel ReasonDiff model for unpaired text-image to video generation, for the first time. Specifically, to analyze the intrinsic connections between the given conditions, we design a VisionNarrator module to leverage the strong reasoning capabilities of a multi-modal large language model (MLLM), and generate a plausible per-frame narrative to recover the whole scene, temporally aligning the unpaired modalities. The VisionNarrator first infers the most likely position of the condition image within the final video, enabling more accurate and context-aware generation. To bridge the reasoning outputs with the generation process, we introduce the AlignFormer module, which treats the condition image as an anchor and predicts the latent representations for the remaining frames. Concretely, AlignFormer employs a Multi-stage Temporal Anchor Attention (MTAA) mechanism that progressively refines latent representations through a cascade of cross-attention layers, effectively injecting reasoning signals into the feature space. The resulting reasoning-enhanced latents are then fused with the condition frame, providing precise, frame-wise control. During the training stage, we will first warm up the whole model using the standard denoising loss, and then add an auxiliary reconstruction loss between the predicted reasoning enhanced latents and the matching ground-truth latents to fine-tune the AlignFormer module individually. In this way, the ReasonDiff model is able to reason out the possible scene from the seemingly unrelated conditions, and generate a video that is realistic and semantically-coherent with both the image and the text. Our contributions are summarized as follows:

- To the best of our knowledge, we for the first time propose to solve the challenging problem of unpaired text-image to video generation.

- To tackle the challenges in the above problem, we propose an MLLM Driven Multi-frame Reasoner, comprising two key components, namely VisionNarrator and AlignFormer, which derives a per-frame narrative that is coherent with the unpaired inputs and predicts latent representations for unseen frames, respectively.

- We design a Reasoning Guided Generative Model to empower the base video generative model with reasoning abilities and propose an end-to-end training procedure under unpaired text-image inputs.

- We conduct extensive experiments and ablation studies to verify the strong reasoning and generating abilities of the proposed ReasonDiff model.

## 2 RELATED WORK

**Video Generative Models.** Diffusion models have become a powerful framework for video synthesis, producing realistic and temporally coherent results. By extending DDPM (Ho et al., 2020) to text/image-to-video tasks, they incorporate temporal dynamics and learn motion priors from large-scale datasets like WebVid (Bain et al., 2021). Flow matching (Lipman et al., 2022) later reframes this as a distribution mapping problem, offering a stronger training objective. Early work such as Video Diffusion Model (Ho et al., 2022) suffers from low resolution problems. Subsequent methods (Blattmann et al., 2023b; Luo et al., 2023; Zhang et al., 2024) leverage spatial-temporal upsampling to enhance video quality.

To achieve a better controllable generation, recent works have studied to add various condition signals (Zhang et al., 2023a; Yin et al., 2023), such as poses (Ma et al., 2024; Zhang et al., 2023b) and structures (Xing et al., 2024a; Esser et al., 2023). Specifically, for (text-)image-to-video generation, Dynamicrafter (Xing et al., 2024b) fuses the condition information with the initial noises and proposes spatial dual-attn transformer module to support more precise conditioning. LTX-Video (Ha-Cohen et al., 2024) seamlessly integrate Video-VAE into denoising transformers, and optimize their interaction for improved efficiency and quality. And more recently, large-scale video generation models such as CogVideoX (Yang et al., 2024) and Wan2.1 (Wan et al., 2025) use DiT as backbone and can generate highly-realistic videos. Nonetheless, these models typically assume that the input text and image are perfectly paired, implicitly relying on the alignment between the two modalities to guide the generation process. In scenarios where the text and image are loosely related or entirely unpaired, such models often fail to reason about their intrinsic connections. This leads to generated videos that lack semantic coherence and exhibit poor visual consistency, limiting their applicability in real-world, weakly supervised settings where perfect alignment is rare.

**Generation with Reasoning.** Recently, with the emergence of ChatGPT (Achiam et al., 2023) and other language models (Liu et al., 2024; GLM et al., 2024), researches on large language models (LLMs) have gained significant momentum. In particular, there has been growing interest in exploring the reasoning capabilities inherent in these models. Some researches (Shum et al., 2023; Zhang et al., 2022; Wei et al., 2022) prove the reasoning abilities can be enhanced through prompt refinement. Zero-shot-CoT (Kojima et al., 2022) achieves performance boost by simply adding *Let's think step by step* before each answer. MM-CoT (Zhang et al., 2023c) incorporates language and vision modalities into a two-stage framework that separates rationale generation and answer inference.

For generative models, LLM has been widely used to infer additional information from the given conditions, such as scene layout (Lian et al., 2023; Long et al., 2024) and object relationship (Li et al., 2024). VQAI (Li et al., 2024) introduces casual reasoning in image generation, and extends the visual question answering tasks to include image as answer. SmartEdit (Huang et al., 2024a) addresses complicated instruction-based image editing problem with the reasoning abilities of LLMs and bidirectional information interactions. And regarding video generation, LayoutGPT (Feng et al., 2024) leverages LLMs to generate detailed scene descriptions along with multiple bounding boxes. Similarly, VideoDirectorGPT (Lin et al., 2023) enhances the controllability of generation by incorporating scene descriptions and layout information produced by LLMs. However, these works do not take the possible complicated interaction between the conditions, *i.e.*, image and text, into consideration, and thus can not handle the situation when the inputs are unpaired.

## 3 METHOD

In this section, we will describe our proposed ReasonDiff method in detail. The overall framework is illustrated in Figure 2. It mainly consists of a base Reasoning Guided Generative Model which is built upon Wan2.1 (Wan et al., 2025), a VisionNarrator and an AlignFormer.

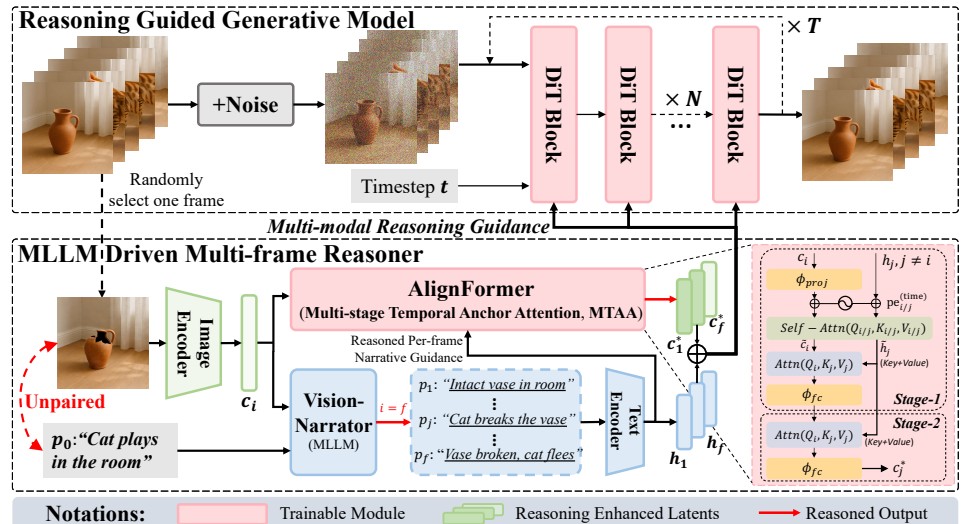

Figure 2: Overview of the ReasonDiff model, which consists of two key components: (1) the *MLLM-Driven Multi-frame Reasoner*, and (2) the *Reasoning-Guided Generative Model*. The generative model operates under the guidance of the multi-modal reasoning results generated by the reasoner.

## 3.1 PRELIMINARY

**Flow matching** Diffusion models have seen widespread applications in AIGC scenarios, and Flow matching (Lipman et al., 2022) has become the standard training objective for existing generation models using DiT as the backbone. Flow matching extends DDPM and learns the mapping between two distributions. Specifically, given data $x_1 \sim q(x)$ and gaussian noise $x_0 \sim \mathcal{N}(0, 1)$, the model is optimized to transform $x_0$ into $x_1$ via predicting the velocity field, *i.e.*,

$$\mathcal{L} = \mathbb{E}_{x_1, x_0 \sim \mathcal{N}(0,1), y, t \sim \mathcal{U}(0,1)} \left[ ||u_\theta(x_t, y, t) - v(x_t)||_2^2 \right], \quad (1)$$

where $t$ is the timestep, $x_t = tx_1 + (1-t)x_0$ is an intermediate noisy latent, $y$ represents an optional conditioning signal, and $u_\theta(\cdot)$ is the prediction model parameterized by $\theta$. $v(x_t)$ is the conditional velocity field, namely,

$$v(x_t) = v(x_t \mid x_1) = x_1 - x_0. \quad (2)$$

In terms of video generation, the widely adopted approach is to treat the video as a sequence of images and perform self-attention along the temporal axis to learn the motion priors. Thus a typical T2I generative model with flow matching objective can be extended to a video generative model after appropriate fine-tuning on video datasets.

**Task** This paper addresses the problem of unpaired text-image to video generation, specifically, generating a sequence of video frames $x \in \mathbb{R}^{ch \times f \times h \times w}$ given a text prompt $p_0$ and an image $y \in \mathbb{R}^{ch \times h \times w}$, such that the output is semantically coherent with both inputs, where $ch$, $f$, $h$ and $w$ represents the channel, frame, height and width, respectively. Importantly, the text and image are unpaired, *i.e.*, the image is not guaranteed to share the same semantic information with the prompt, nor is it necessarily the first frame of the target video. As a result, directly applying existing video generative models often yields suboptimal results due to their limited ability to reason over loosely aligned or entirely unpaired multi-modal inputs.

## 3.2 VISIONNARRATOR

Given unpaired image and text conditions, existing video generative models struggle to infer a coherent narrative, often failing to generate semantically consistent videos. To address this limitation, we propose VisionNarrator and leverage the strong reasoning capabilities of a multi-modal large language model to analyze the underlying connections between the two modalities. Specifically, the MLLM anchors the condition image to a plausible frame within the generated video and constructs

a frame-by-frame narrative around this anchor, guided by the combined context of vision and text. To achieve this, we design the following prompt, *i.e.*,

- *You are given an unpaired image and text prompt. Your task is to infer a coherent scene that logically connects both inputs, even if they appear unrelated.*

- *Estimate the most likely position of the image within an `f`-frame video. Then, generate descriptions with rich information for each of the `f` frames that together form a consistent video script.*

- *Respond strictly in the following format: {"position": `j`, "descriptions": [description for frame 1, ..., description for frame `f`]}. Do not include any additional explanations, comments, or formatting.*

This serves as a general instruction and ensures that the output will adhere to the specified format. In practice, we apply in-context learning (Liu et al., 2021) techniques to further stabilize the results.

Consider the example in Figure 1, where the input image shows a broken vase and the text prompt describes a cat playing in the room. The reasoning results produced by the VisionNarrator are presented in Figure 3, with the deduced prompts aligned to their corresponding frames, each highlighted in a different color. We select some key prompts in the generated storyline, namely, *Intact vase → Cat enters the room, breaking the vase → Vase broken, cat flees*, together with their cor-

{ **Position**: 81, **Descriptions**: [ "An intact vase standing on the floor", ... , "A cat enters the room, breaking the vase", ... , "Vase broken, cat flees"] }

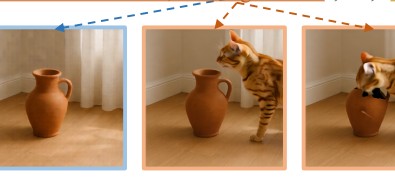

Figure 3: Reasoning results generated by the VisionNarrator. The conditions are the same as in Figure 1. We select some of the generated key frames and connect them with the corresponding prompts using different colors.

responding video frames. We can see that the narrative forms a plausible deduction based on the given unpaired inputs and naturally positions the condition image as the final frame. As illustrated, the VisionNarrator effectively infers the underlying narrative, that the playful cat causes the vase to break, and generates frame descriptions that are semantically aligned with this inferred storyline. The resulting per-frame script, along with the predicted anchor position of the condition image, is then passed to AlignFormer for further processing. It is important to note that although the Vision-Narrator does not directly participate in the video generation process, its reasoning capabilities are essential for constructing each frame's context and ensuring the overall coherence of the video.

### 3.3 ALIGNFORMER

To bridge the gap between the VisionNarrator and the base video generative model, we introduce the AlignFormer module, a Transformer-style architecture that aligns the high-level reasoning outputs with the frame-wise latent features.

The AlignFormer module takes three inputs: (1) the anchor feature $c_i$ extracted from the condition frame, (2) its inferred position $i$ within the target video, and (3) the reasoned per-frame narrative embedding $h = \{h_j\}_{1 \le j \le f}$. It then outputs a sequence of reasoning enhanced latent features, *i.e.*, $c^* = \{c_j^*\}_{1 \le j \le f}$, corresponding to each frame. The module structure is illustrated in the lower-right region of Figure 2.

In particular, the module employs a Multi-stage Temporal Anchor Attention (MTAA) mechanism to progressively synthesize each frame's latent representation by integrating the anchor feature with temporally structured semantic guidance derived from per-frame narratives. This is achieved through a two-stage cross-attention process: the first stage is designed to capture coarse temporal dependencies, while the second stage refines the representations with finer contextual alignment. At each stage, the anchor feature acts as the *Query*, while the corresponding narrative embeddings are projected to form the *Key* and *Value* representations, *i.e.*,

$$\tilde{c}_i = \phi_{\text{proj}} \left( \text{Flatten}(c_i) \right) + \text{pe}_i^{(\text{time})}, \ \tilde{h}_j = h_j + \text{pe}_j^{(\text{time})}, \tag{3}$$

$$Q_i = W_Q \tilde{c}_i, \ K_j = W_K \tilde{h}_j, \ V_j = W_V \tilde{h}_j, \tag{4}$$

$$c_j^* = \text{Attn}(Q_i, K_j, V_j) = \text{Softmax}\left( Q_i K_j^T / \sqrt{d} \right) V_j, \tag{5}$$

where $j \neq i$ is the index for the predicted latent feature, $W_Q$, $W_K$ and $W_V$ are the projection matrices for *Query*, *Key* and *Value*, and $\text{pe}_{i/j}^{(\text{time})}$ represents positional embedding along the temporal axis. This MTAA mechanism adopted in AlignFormer helps to effectively align the visual and textual representations, enabling the transfer of high-level reasoning. The resulting reasoning-enhanced latents, denoted as $c^*$, together with the prompt embedding $h$, are then fused with the condition frame $c_i$ to serve as guidance throughout the denoising steps of the generation process. Compared to directly injecting the multi-frame prompts without AlignFormer, our ReasonDiff model, equipped with this module, achieves noticeably better generation quality and temporal coherence (please see Section 4.3 for more details).

### 3.4 TRAINING PROCEDURE

In this subsection, we describe the training procedure of the proposed ReasonDiff model. A key challenge lies in the scarcity of multi-modal datasets featuring unpaired text-image conditions. Since most existing works assume paired inputs in video generation, there is currently no available dataset that can serve as direct ground truth for training ReasonDiff. To address this, we leverage the reasoning capabilities of a pre-trained multi-modal large language model. The VisionNarrator is set frozen and functions solely as a reasoning module to bridge the semantic gap between unpaired conditions. Consequently, the only trainable components in our framework are the base video generative model and the newly introduced AlignFormer module. This reformulates our training task into a conditional video generation problem, where the model reconstructs video clip $x \in \mathbb{R}^{b \times ch \times f \times h \times w}$ given a randomly selected condition frame indexed $i \in \{1, \ldots, f\}$ and a corresponding sequence of per-frame prompt embeddings $h \in \mathbb{R}^{f \times l \times d}$, where $b$, $l$ and $d$ represents the batch size, context length and embedding dimension within the text encoder, respectively. Moreover, to better simulate the unpaired condition scenario during training, we increase the temporal interval between selected frames in each video clip, ensuring that the condition frame appears less correlated with the surrounding contents. In this way, we can effectively train our model based on existing video datasets.

Concretely, we use video data from WebVid dataset (Bain et al., 2021), sampling frames at 0.2 second intervals. For each frame, we generate a corresponding caption using LLaMA-3.2-11B-Vision-Instruct (Grattafiori et al., 2024). During training, a random frame is selected as the condition frame, and the model is trained to reconstruct the entire video based on this frame and the corresponding per-frame textual descriptions.

Our training procedure consists of two stages. In the first stage, we jointly train the pre-trained base video generative model and the AlignFormer using a standard denoising loss. This phase serves to initialize the newly added module and align it with the flow-based generation process. In the second stage, we introduce an auxiliary reconstruction loss between the predicted latent features $c^*$ and the ground-truth latent features $c$, encouraging the model to better align generated representations with the original video contents. Specifically, the second-stage loss is defined as follows:

$$\mathcal{L} = \mathbb{E}_{x_1, x_0 \sim \mathcal{N}(0,1), h, t \in \mathcal{U}(0,1), c} \left[ ||u_\theta(x_t, h, c^*) - v(x_t)||_2^2 + \beta \cdot ||c^* - c||_2^2 \right] \tag{6}$$

where $\beta$ is a hyper-parameter controlling the weight of the auxiliary loss, and we keep the parameters of the base generative model fixed to fine-tune the AlignFormer alone. In practice, we set $\beta = 0.2$.

## 4 EXPERIMENT

In this section, we first detail on the specific settings of the proposed ReasonDiff, and conduct extensive quantitative and qualitative experiments under unpaired text-image conditions. We further conduct some ablation studies to verify the effectiveness of each module.

| Dataset | Model | Imaging Quality(↑) | Motion Smooth(↑) | Dynamic Degree(↑) | CLIP Score (Text)(↑) | CLIP Score (Image)(↑) | User Rank(↓) |
|---|---|---|---|---|---|---|---|
| **ActivityNet** (Self-constructed) | Dynamicrafter | 0.492 | 0.979 | 0.484 | 0.202 | 0.508 | 2.871 |
| | LTX-Video | 0.398 | 0.977 | 0.734 | 0.211 | **0.544** | 3.307 |
| | CogVideoX | 0.507 | 0.949 | 0.872 | 0.197 | 0.537 | 4.384 |
| | Wan2.1 | 0.512 | 0.980 | 0.810 | 0.224 | 0.518 | 2.692 |
| | **ReasonDiff** | **0.528** | **0.986** | **0.936** | **0.261** | 0.528 | **1.743** |
| **MSR-VTT** (Public-General-purpose) | Dynamicrafter | 0.517 | 0.984 | 0.440 | 0.201 | 0.526 | 3.179 |
| | LTX-Video | 0.406 | **0.986** | **0.695** | 0.206 | **0.588** | 4.051 |
| | CogVideo | 0.552 | 0.970 | 0.688 | 0.177 | 0.572 | 3.256 |
| | Wan2.1 | 0.560 | 0.962 | 0.665 | 0.191 | 0.552 | 2.743 |
| | **ReasonDiff** | **0.571** | 0.984 | 0.673 | **0.214** | 0.572 | **1.769** |

Table 1: Quantitative comparison between ReasonDiff and the baselines. The top and second top performances have been bolded or underlined respectively. Complete table with standard errors can be found in Appendix C.3.

## 4.1 EXPERIMENTAL SETUP

Since our work targets video generation under unpaired text-image conditions, we construct a custom evaluation dataset to align with this objective. Specifically, we randomly sample 500 videos from ActivityNet (Caba Heilbron et al., 2015) and extract a 16-frame clip from each. For each clip, we select either the first or the last frame as the condition image and use LLaMA-3.2-11B-Vision-Instruct to generate a caption for the opposite end (the last or first frame, respectively) as the prompt. This setup ensures a temporal separation between the two conditions, thereby simulating an unpaired scenario. Importantly, the model will perform generation without access to the frame index or the relative temporal position between the given image and text. In addition, we incorporate a public general-purpose dataset MSR-VTT (Xu et al., 2016), which contains paired conditions, to enable a more comprehensive comparison and to further assess the general generative ability of the method.

We compare our proposed ReasonDiff model with the following baselines: (1)Dynamicrafter (Xing et al., 2024b), (2)LTX-Video-2B (HaCohen et al., 2024), (3)CogVideoX-1.5-5B (Yang et al., 2024) and (4)Wan2.1-I2V-14B (Wan et al., 2025), which are latest works on video generation that have been open-sourced and achieve good performances. We have employed six metrics, namely *Imaging Quality*, *Motion Smooth*, *Dynamic Degree*, *CLIP Score (Text/Image)* and *User Rank*. Note that the first three metrics are general evaluation criteria supported by VBench (Huang et al., 2024b), and the two *CLIP Score*s quantify the semantic alignment between the generated video and the relevant conditions (text/image). Implementation and evaluation details can be found in Appendix A and B.

## 4.2 MAIN RESULTS

We conduct both quantitative and qualitative experiments, with the results presented in Table 1 and Figure 4. In quantitative evaluation, ReasonDiff shows competitive performance compared to state-of-the-art baselines on the general purpose dataset MSR-VTT (*i.e.*, with paired conditions), while achieving top results across all metrics except for *CLIP Score (Image)* on the self-constructed ActivityNet dataset that simulates unpaired conditions. It is important to note that ReasonDiff significantly outperforms all baselines in *CLIP Score (Text)* and *User Rank* on ActivityNet, which are the two most critical metrics for evaluating model performance with unpaired image and text. Specifically, ReasonDiff exceeds the best-performing baseline, Wan2.1, by 16.5% in *CLIP Score (Text)*, and by 0.949 in *User Rank*. As for *CLIP Score (Image)*, all methods show comparable performances, with scores hovering around 0.5. This is consistent with the observation in Figure 1, where baseline models tend to rely heavily on the input image when confronted with unpaired conditions. As a result, their *CLIP Score (Image)*s remain relatively high, as well as ReasonDiff's, making this metric less discriminative for unpaired settings. However, when jointly considering both *CLIP Score* metrics, it becomes evident that the baselines lack the capability to integrate and reason across modalities. They focus primarily on visual conditions while neglecting the semantic guidance, leading to poor alignment with the textual input and ultimately less coherent video generation. In contrast, ReasonDiff achieves high scores across both metrics, reflecting its robust reasoning capabilities to generate semantically rich and coherent videos even under unpaired text-image conditions.

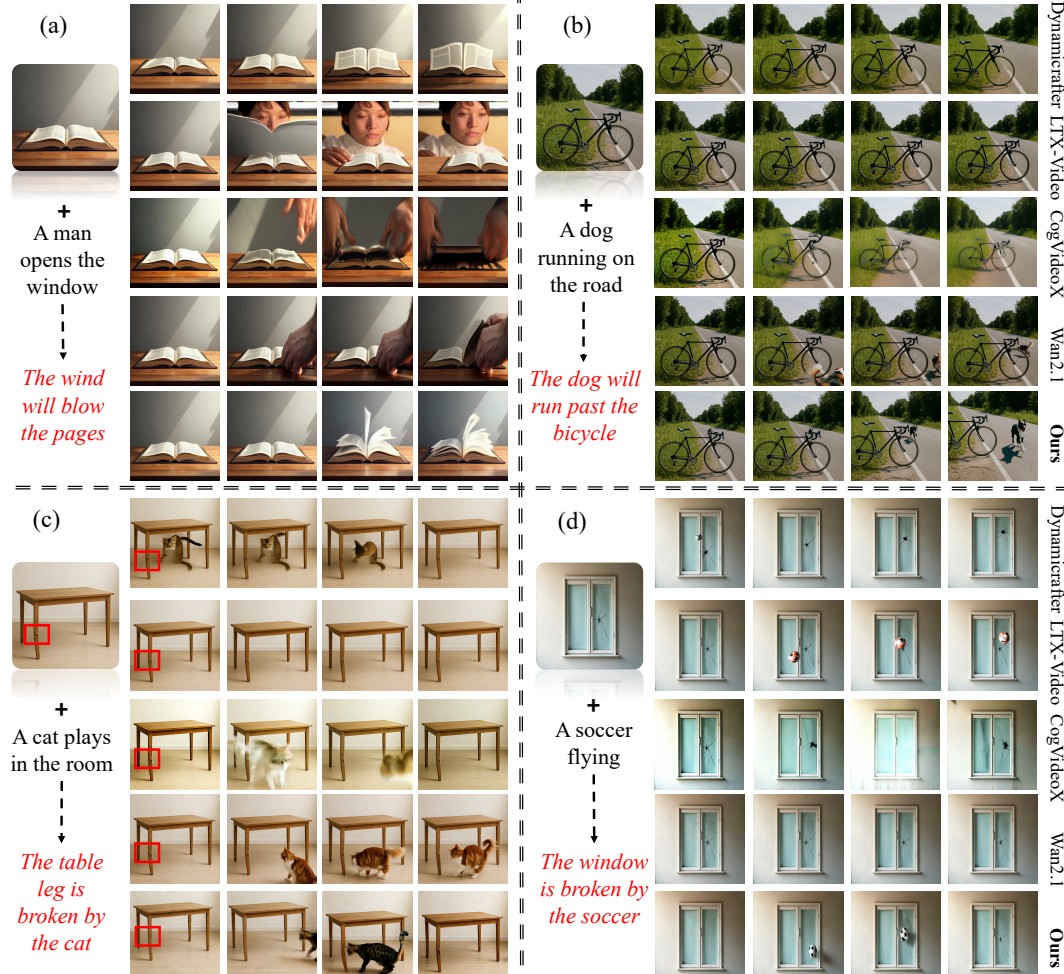

Figure 4: Quantitative comparison between ReasonDiff and the baselines. We select several intermediate frames for the convenience of presentation. More generated samples can be found in Appendix C.3 and the supplementary materials.

In qualitative experiments, we evaluate ReasonDiff and the other baselines using the same unpaired condition image and text. The results are presented in Figure 4. As shown, baseline models generally struggle to infer meaningful connections between the unpaired conditions, resulting in two major types of failure cases: (1) confusing content, where the generated frames contain entangled visual elements due to conflicting signals from the unpaired inputs (e.g., outputs from CogVideoX in Figure 4(a), which shows bizarre interaction of the *hand* and *book*); and (2) incoherence with the unpaired conditions, where the expected motions implied by the prompt do not occur (e.g., outputs from Dynamicrafter in Figure 4(b), LTX-Video in Figure 4(c) and Wan2.1 in Figure 4(d), which fail to depict *dog running past the bicycle*, *cat breaks the table leg* or *soccer breaks the window*, respectively). In contrast, ReasonDiff produces videos that closely follow the semantic intent of both the image and the text, demonstrating its superior ability to reason over unpaired multi-modal inputs.

## 4.3 ABLATION STUDIES

In this section, we evaluate the effectiveness of the proposed modules through comprehensive ablation studies. We design four variants of the full ReasonDiff model, namely, (1) *w/o. Aux. loss*, which removes the second training stage and disables the auxiliary reconstruction loss; (2) *w/o. Multi. prompt*, which uses only the single user-provided prompt without the per-frame narratives from VisionNarrator; (3) *w/o. Enhanced latents*, which disables the enhanced latents, relying solely

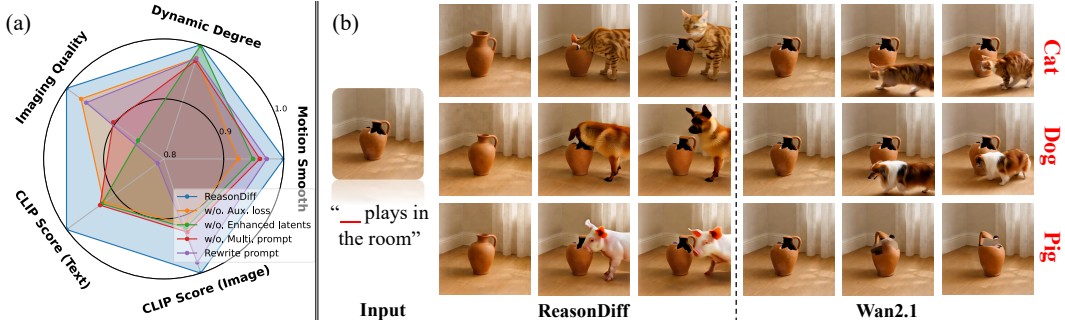

Figure 5: (a) Ablation studies on four variants of ReasonDiff. All metrics are reported as ratios relative to the full model. (b) Comparison between ReasonDiff and Wan2.1 given the same condition image and varying text prompts, *i.e.*, *Cat/Dog/Pig plays in the room* (causing the vase to be broken).

on narrative guidance; and (4) *Rewrite prompt*, which employs an MLLM to rewrite the text prompt based on the given condition image, thereby assisting the reasoning process of the base model. For variant (1)-(3), the training procedure of ReasonDiff is modified to isolate the contribution of each module, whereas for variant (4), the rewritten prompt is directly supplied while keeping the pre-trained base model unchanged. We compare the performance of ReasonDiff against these four variants on ActivityNet, and present the results in Figure 5(a).

Specifically, disabling *Enhanced latents* results in a substantial degradation in both *Imaging Quality* and *CLIP Score (Text)*, underscoring its critical role in providing strong guidance under unpaired conditions. In contrast, removing *Aux. loss* or *Multi. prompt* leads to notable declines in *Motion Smoothness* and *Dynamic Degree*. The *Rewrite prompt* variant achieves relatively high scores on *CLIP Score (Image)*, but at the cost of a severe reduction in *CLIP Score (Text)*. Overall, ReasonDiff consistently outperforms all ablated variants across every metric, demonstrating the necessity and effectiveness of each proposed component.

### 4.4 APPLICATION: PROMPT-DRIVEN CUSTOMIZATION

In this section, we show one possible application of ReasonDiff as *Prompt-driven Customization*. In video generation, there is often no single ground-truth answer due to the inherent uncertainty of the task. For instance, recovering a broken scene from an image of a shattered vase alone is ambiguous. However, when jointly conditioned on a corresponding prompt, such as *cat plays in the room*, the model can reason that the cat likely causes the vase to break. Since different prompts can imply different plausible causes, the model must be capable of customizing the generated video accordingly. As shown in Figure 5(b), we use the same condition image while varying the subject in the prompt (e.g., cat, dog, or pig), and compare the outputs of ReasonDiff and Wan2.1. We can observe that Wan2.1 fail to uncover the connections between the two modalities, even generating confusing scenes under the subject *pig*. In contrast, ReasonDiff successfully infers the correct relationship between the unpaired input image and text, producing videos that align with the varying prompts and keep narrative coherence.

## 5 CONCLUSION

In this work, we for the first time propose to solve unpaired text-image to video generation and present a novel ReasonDiff model. Unlike existing approaches that often produce visually confusing videos that mix multiple objects incoherently or failing to uncover the intrinsic connections, ReasonDiff is designed to reason over both unpaired modalities simultaneously. To tackle the challenges, we introduce two key components, *i.e.*, VisionNarrator and AlignFormer. VisionNarrator extracts a per-frame narrative based on the unpaired inputs, while AlignFormer predicts reasoning-enhanced latents to guide the base video generator. Together, these modules enable the generation of videos from unpaired inputs that achieve both photorealism and semantic coherence.

## 6 REPRODUCIBILITY STATEMENT

We provide comprehensive implementation details for both training and inference, including pseudocode in Appendix A and Appendix B, to enhance the reproducibility of our approach. In addition, we will release the code and pre-trained weights to facilitate further research and exploration.

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

## A    IMPLEMENTATION

### A.1    MODEL AND HYPER-PARAMETERS

Our base generative model is built upon Wan2.1 (Wan et al., 2025), which generates a photorealistic video from a static image and a corresponding prompt. Specifically, Wan2.1 is built on Diffusion Transformer (DiT), and adopts mask mechanism and decoupled cross attention. For more details, please refer to their official report. The version we use is Wan2.1-I2V-14B-480P[1]. For Vision-Narrator, we use LLaMA-3.2-11B-Vision-Instruct[2] (Grattafiori et al., 2024) to generate a plausible per-frame narrative for the training data, while in inference, we utilize the state-of-the-art MLLM gpt-4o (Hurst et al., 2024) model. For AlignFormer, the exact architecture of which has been illustrated in Figure 2 in the main paper. We set one self-attention layer and two cross-attention layers for the integration of reasoning capabilities, with fully-connected layer following each cross-attention layer, and fix the number of heads in a multi-head attention layer to 8.

### A.2    TRAINING AND INFERENCE

In this section, we give the details on the implementation for training and inference. During training, the amount of data we use is approximately 10k, which is processed from WebVid dataset (Bain et al., 2021). We set batch size to 8, sampling frames at 0.2 second intervals, and fix the number of frames and the frame resolution to 33 and $512 \times 512$, respectively. We add LoRA layers to the base Wan2.1 model, while randomly initializing the AlignFormer module, and we choose the AdamW optimzer with a learning rate of 1e-5. Furthermore, we conduct one epoch of training in the first stage, and an additional epoch in the second stage, with the balancing parameter $\beta$ set to 0.2. All training is conducted on one A100 80G GPU.

During inference, we fix the length of the generated video to 81 frames. Moreover, We set the number of timesteps in the denoising process to 50, and fix the scale of the classifier-free guidance to 5.0.

Additionally, we give the overall implementation of the training procedure in Algorithm 1, which consists of two stages, where we first train the base video generative model and the newly added AlignFormer module as a whole, and then we introduce an auxiliary reconstruction loss to fine-tune the AlignFormer individually. Note that we have simplified the *MultiFrameReasoning* function, *i.e.*, the role of AlignFormer module, in the algorithm. The notations are the same as in the main paper.

## B    EVALUATION DETAILS

### B.1    BASELINES

We compare our method with the following baselines, which are the latest works for video generation that achieve good performances, specifically,

1. Dynamicrafter (Xing et al., 2024b)[3]: Dynamicrafter studies the animation of open-domain images by using a query transformer to project the input image into a text-aligned rich context representation space and fuse the initial image in the diffusion process.

2. LTX-Video-2B (HaCohen et al., 2024)[4]: LTX-Video seamlessly integrate the video-vae and denoising transformer, jointly optimizing their interaction for improved efficiency and quality. Notably, it achieves faster-than-real-time generation.

3. CogVideoX1.5-5B (Yang et al., 2024)[5]: CogVideoX address the problem of the generation of long videos that are coherent with the conditions. They introduce a 3D-VAE to compress

---

[1] https://huggingface.co/Wan-AI/Wan2.1-I2V-14B-480P
[2] https://huggingface.co/meta-llama/Llama-3.2-11B-Vision-Instruct
[3] https://huggingface.co/Doubiiu/DynamiCrafter/blob/main/model.ckpt
[4] https://huggingface.co/Lightricks/LTX-Video/blob/main/ltxv-2b-0.9.6-distilled-04-25.safetensors
[5] https://huggingface.co/zai-org/CogVideoX1.5-5B-I2V

---

**Algorithm 1** Training procedure of ReasonDiff.

---

1: **Input**: Data $= \{(x_1 \in \mathbb{R}^{b \times c \times f \times h \times w}, h \in \mathbb{R}^{f \times l \times d})\}_{N_D}$
2: **Parameter**: $\theta_R$: parameters for ReasonDiff, $\theta_V$:parameters for base video generative model.
3: **function** MULTIFRAMEREASONING($c_i, h, f$)
4:     $c^* \leftarrow$ repeat($c_i, f$).
5:     $c^* \leftarrow$ Self-Attn($c^*$), $h \leftarrow$ Self-Attn(h).
6:     $c^* \leftarrow$ Cross-Attn(query $= c^*$, key $= h$, value $= h$).
7:     $c^* \leftarrow$ MLP($c^*$) $+ c^*$.
8:     $c_i^* \leftarrow c_i$.                              ▷ Use anchor $c_i$ to replace $i^{th}$ element of $c^*$.
9:     **return** $c^*$.
10: **end function**
    BEGIN MAIN FUNCTION:
11: Initialize $f$, $\beta$ as the number of frames and the weight for the auxiliary loss.
12: **for** $\theta \in [\{\theta_R, \theta_V\}, \{\theta_R\}]$ **do**
13:     **repeat**
14:         Forward a batch of data $\{(x_1, h)\}_{N_{\text{batch}}}$.
15:         Sample $x_0 \sim \mathcal{N}(0, 1)$.
16:         $x_t \leftarrow tx_1 + (1 - t)x_0$.
17:         Randomly select the condition index $i \in \{1, \ldots, f\}$.
18:         $c \leftarrow$ VIDEOFRAMEENCODER($x_1$).
19:         $c^* \leftarrow$ MULTIFRAMEREASONING($c_i, h, f$).
20:         loss $\leftarrow \|u_{\theta_V}(x_t, h, c^*) - v(x_t)\|_2^2$.
21:         **if** $\theta = \{\theta_R\}$ **then**                        ▷ Second stage training.
22:             loss $\leftarrow$ loss $+ \beta \cdot$ MSE($c, c^*$)
23:         **end if**
24:         $\theta \leftarrow \arg\min_\theta(\text{loss})$.
25:     **until** One Epoch Ends
26: **end for**

---

    the videos, and further propose expert adaptive LayerNorm and progressive training to improve video quality.

    4. Wan2.1-I2V-14B-480P (Wan et al., 2025)[6]: Wan2.1 is a comprehensive video foundation model comprising a novel spatial-temporal variational autoencoder and scalable pre-training strategies. It is built upon DiT architecture with parameters on the scale of billion.

## B.2 DATASET

Since our work focuses on video generation under unpaired text-image conditions, we construct a custom evaluation dataset tailored to this setting. Specifically, we randomly sample 500 videos from ActivityNet (Caba Heilbron et al., 2015) and extract a 16-frame clip from each. For every clip, we select either the first or the last frame as the condition image and use LLaMA-3.2-11B-Vision-Instruct to generate a caption for the opposite end (*i.e.*, the last or first frame, respectively) to serve as the text prompt. This design introduces temporal separation between the visual and textual conditions, effectively simulating an unpaired scenario. Additionally, we incorporate the public general-purpose dataset MSR-VTT (Xu et al., 2016) by directly using its validation set, which contains approximately 500 video clips. For each clip, we use the first frame as the condition image.

## B.3 METRICS

In this section, we elaborate more on the details of the metrics employed in the quantitative experiment, *i.e.*, *Imaging Quality*, *Motion Smooth*, *Dynamic Degree*, *CLIP Score (Image/Text)* and *User Rank*. The first three metrics, which are universal evaluation criteria for general videos, are supported by VBench (Huang et al., 2024b), an open-sourced evaluation benchmark on video domain. As indicated by the name, *Imaging Quality* measures the aesthetic level of the generated frames, and *Motion Smooth* judges whether the movements and transformations in the video are photorealistic,

---

[6]https://huggingface.co/Wan-AI/Wan2.1-I2V-14B-480P

while *Dynamic Degree* favors those with more motion dynamics. The rest of the metrics are as follows:

**CLIP Score (Image/Text)** We employ a CLIP encoder[7] (Radford et al., 2021) to extract features from the generated video frames, the condition image and the text prompt, and compute the average cosine similarity as the final evaluation metric. Specifically, for *CLIP Score (Text)*, we utilize LLaMA-3.2-11B-Vision-Instruct to generate captions for the ground-truth video frames, then calculate the similarity between each caption and its corresponding generated frame. This approach ensures that higher scores reflect stronger semantic alignment between the generated video and the ground-truth sequence.

**User Rank** In the user study, each participants are given multiple randomly-chosen sets of questions, each containing unpaired input image and text, with five corresponding and randomly-arranged video samples generated from ReasonDiff and the baseline models, which should be ranked based on their coherence with the input conditions, intuitive feeling and overall quality, etc. Specifically, we ask 20 participants to each answer ten questions, and average the scores for each model respectively as the final *User Rank* for the two datasets.

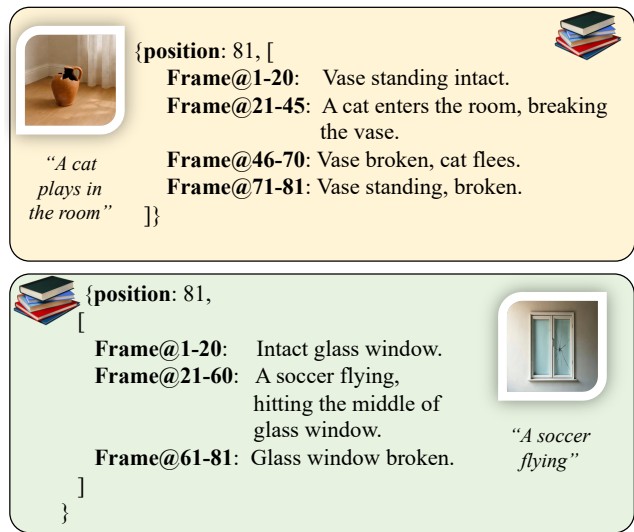

Figure 6: Reasoning examples of the proposed VisionNarrator, where we give the deduced position of the condition image and the detailed per-frame narrative following the specific format.

## C ADDITIONAL EXPERIMENTS

### C.1 REASONING EXAMPLE OF VISIONNARRATOR

In this section, we give some detailed reasoning examples of VisionNarrator, and present the results in Figure 6. For instance, given the image of *a broken vase* and the text prompt of *A cat plays in the room*, the VisionNarrator divides the whole scene into four clips: (1) Frame@1-20, with the vase intact at first; (2) Frame@21-45, with the cat entering the room and breaking the vase; (3) Frame@46-70, with the cat fleeing and lastly, (4) Frame@71-81, with the vase already broken, corresponding to the given condition image. From the results we can observe that the VisionNarrator is capable of reasoning out complicated connections and generate a semantically coherent per-frame narrative to guide the generation process.

---

[7]https://huggingface.co/openai/clip-vit-large-patch14

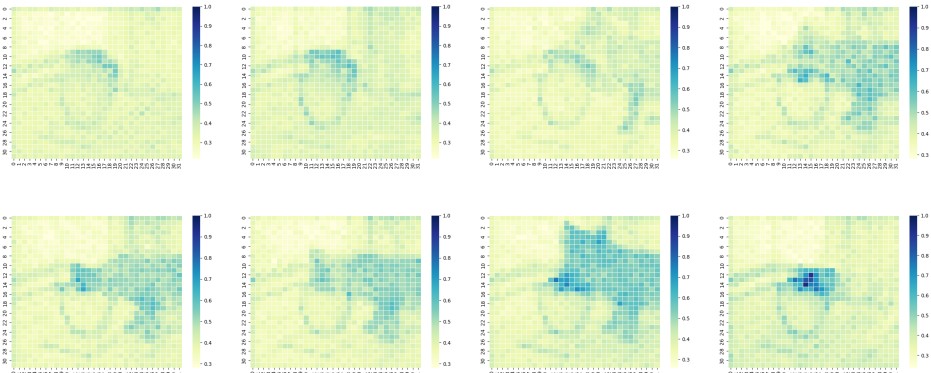

Figure 7: Attention map of the *vase-cat* example in Figure 8(c) (last line, the generated sample from ReasonDiff). Specifically, we compute the attention scores between the CLIP-encoded embedding of the word *breaking* and the hidden states of the video frames in the second transformer block, and illustrate them in time order. The deeper the color, the higher the attention score.

## C.2 ATTENTION-MAP ILLUSTRATION

In this section, we conduct an additional experiment to show that ReasonDiff can successfully reason out the temporal connections and generate a video coherently, rather than just combining the two modalities. Specifically, considering the *vase-cat* example, we compute the attention scores between the CLIP-encoded embedding of the word *breaking* and the hidden states of the video frames in the second transformer block. These attention maps are then averaged across channels to obtain a frame-wise attention distribution. The results are illustrated in Figure 7. From the results we can observe that, the areas with high attention scores follow the interaction of the vase and the cat, with little attention scores at first (when the cat hasn't entered the room). This highlights the model's ability to infer dynamic causal relationships from unpaired text-image inputs.

## C.3 COMPLETE TABLES AND ADDITIONAL QUALITATIVE COMPARISON

In this section, we provide complete tables with standard deviations in Table 2 and Table 3, corresponding to the quantitative comparisons on the self-constructed ActivityNet dataset and the public general-purpose MSR-VTT dataset, respectively. And we give additional qualitative comparisons in Figure 8. As illustrated, we can see that ReasonDiff consistently outperforms baseline models, which often exhibit visual inconsistency, such as object mixing, semantic misalignment, where one of the input conditions is ignored, or failing to identify the possible connections between the multi-modal conditions. In contrast, ReasonDiff generates more coherent and semantically aligned video content across diverse scenarios.

## D  DISCUSSIONS

Although video generation under unpaired text–image conditions has not yet been explored in the literature, it holds significant real-world relevance. On the one hand, user-provided conditions cannot be assumed to be perfectly aligned, leading existing approaches to perform suboptimally as they generally fail to reason across modalities. On the other hand, in tasks that inherently require reasoning, users may deliberately provide unpaired conditions—for example, to recover past scenes or to predict future ones. Consequently, the reasoning capability introduced by our proposed ReasonDiff is essential, enabling more robust and contextually coherent video generation.

However, our method may have a few limitations. Since it builds on Wan2.1 as the base video generative model, it inherits common shortcomings of existing generative approaches. First, it struggles with generating intricate structures like human poses or complicated interactions between objects like shaking hands or kicking football, which are particularly difficult to maintain consistently across frames. Second, our generated videos are still limited in duration, restricting the complexity of motion. Current video generative models generally support only short video clips (typically less than

| Model | Imaging Quality(↑) | Motion Smooth(↑) | Dynamic Degree(↑) | CLIP Score (Text)(↑) | CLIP Score (Image)(↑) | User Rank(↓) |
|---|---|---|---|---|---|---|
| Dynamicrafter | 0.492 ±0.111 | 0.979 ±0.019 | 0.484 ±0.499 | 0.202 ±0.057 | 0.508 ±0.087 | 2.871 ±1.239 |
| LTX-Video | 0.398 ±0.081 | 0.977 ±0.008 | 0.734 ±0.442 | 0.211 ±0.051 | **0.544** ±0.084 | 3.307 ±1.259 |
| CogVideoX | 0.507 ±0.086 | 0.949 ±0.023 | 0.872 ±0.089 | 0.197 ±0.039 | 0.537 ±0.078 | 4.384 ±1.041 |
| Wan2.1 | 0.512 ±0.103 | 0.980 ±0.023 | 0.810 ±0.280 | 0.224 ±0.056 | 0.518 ±0.079 | 2.692 ±1.079 |
| **ReasonDiff** | **0.528** ±0.106 | **0.986** ±0.048 | **0.936** ±0.244 | **0.261** ±0.061 | 0.528 ±0.082 | **1.743** ±1.044 |

Table 2: Complete table with standard deviation for the quantitative comparison between ReasonDiff and the baselines on the self-constructed ActivityNet dataset that simulates unpaired settings. The top and second top performances have been bolded or underlined respectively.

| Model | Imaging Quality(↑) | Motion Smooth(↑) | Dynamic Degree(↑) | CLIP Score (Text)(↑) | CLIP Score (Image)(↑) | User Rank(↓) |
|---|---|---|---|---|---|---|
| Dynamicrafter | 0.517 ±0.123 | 0.984 ±0.017 | 0.440 ±0.496 | 0.201 ±0.043 | 0.526 ±0.091 | 3.179 ±1.189 |
| LTX-Video | 0.406 ±0.087 | **0.986** ±0.010 | **0.695** ±0.460 | 0.206 ±0.037 | **0.588** ±0.078 | 4.051 ±0.971 |
| CogVideo | 0.552 ±0.082 | 0.970 ±0.015 | 0.688 ±0.323 | 0.177 ±0.025 | 0.572 ±0.059 | 3.256 ±1.481 |
| Wan2.1 | 0.560 ±0.111 | 0.962 ±0.023 | 0.665 ±0.183 | 0.191 ±0.036 | 0.552 ±0.075 | 2.743 ±1.140 |
| **ReasonDiff** | **0.571** ±0.109 | 0.984 ±0.028 | 0.673 ±0.470 | **0.214** ±0.044 | 0.572 ±0.092 | **1.769** ±1.245 |

Table 3: Complete table with standard deviation for the quantitative comparison between ReasonDiff and the baselines on the public and general-purpose MSR-VTT dataset. The top and second top performances have been bolded or underlined respectively.

10 seconds) and often fail to maintain temporal consistency as video length increases. One potential direction is to generate longer content auto-regressively by concatenating consistent short clips together. However, as these limitations are beyond the primary scope of unpaired text-image to video generation, which is how to reason out the intrinsic connections and generate videos that are coherent with both the modalities, we leave them for future exploration.

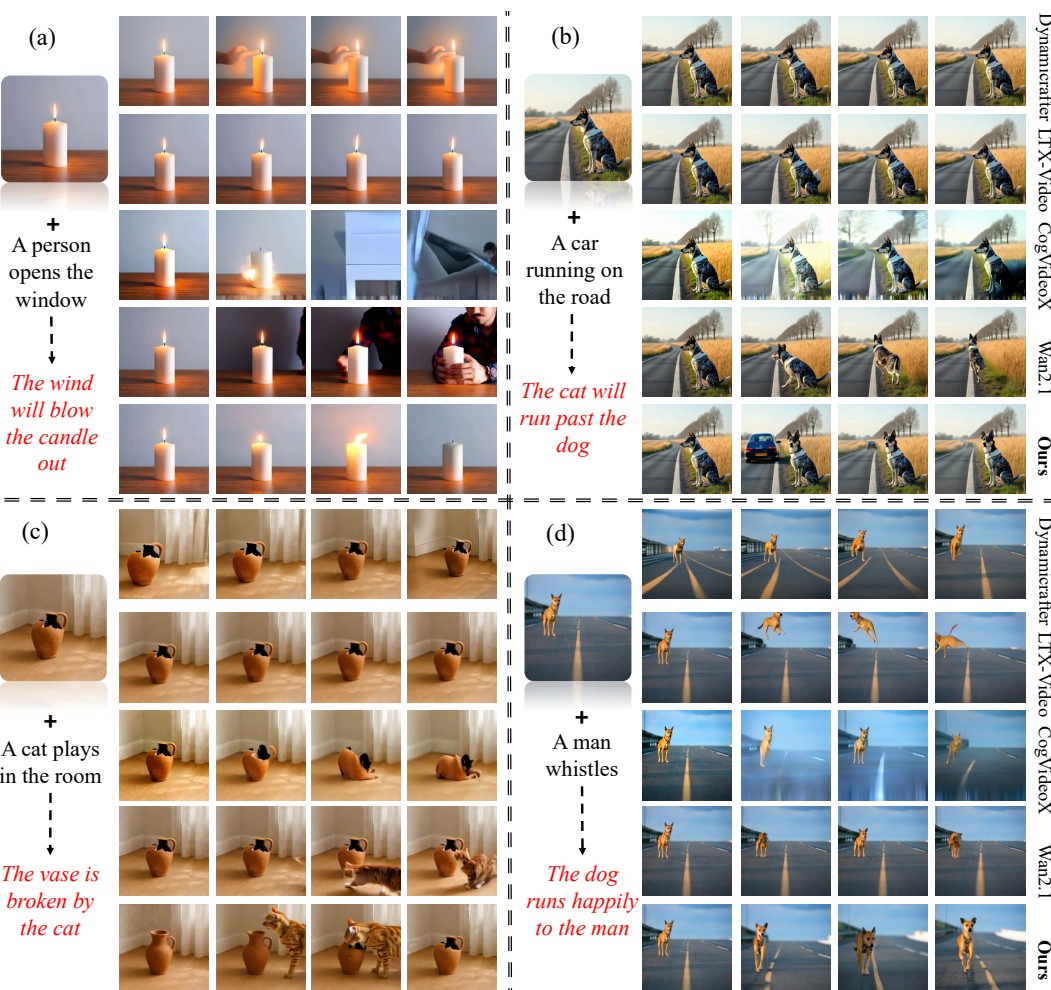

Figure 8: Additional qualitative comparison between ReasonDiff and the baselines. We select several intermediate frames for the convenience of presentation.

