# OpenReview forum: "Reasoning Diffusion for Unpaired Text-Image to Video Generation"
_ICLR.cc/2026/Conference — ICLR 2026 Conference Withdrawn Submission_

### Official Review · Reviewer_rAkg · 2025-10-25

**Soundness:** 3
**Presentation:** 2
**Contribution:** 2
**Rating:** 2
**Confidence:** 4

**Summary:**

This paper introduced a new task, "unpaired text-image to video generation." , and showed why and how current models struggled with it. The author introduced ReasonDiff, which leverages a VisionNarrator to decompose the given prompt across frames and an AlignFormer to replace the original self-attention module for frame-wise alignment. Experiments demonstrated the effectiveness of ReasonDiff on this task.

**Strengths:**

1.	The proposed task and method are novel and well-motivated.
2.	The experiments and ablation studies are thorough and well-designed.
3.	The qualitative results are convincing.

**Weaknesses:**

1. The proposed task, “unpaired text–image to video generation,” is conceptually questionable. Many of the claimed challenges for “unpaired text–image pairs” can be addressed using existing techniques without retraining. The author stated two scenarios:
    1. "The semantic information carried by the text and image may occur at different timestamps."  This can be handled simply by providing **more precise** temporal prompts (e.g., “The cat walks in, and plays with the vase afterward” ). Commercial I2V models often include prompt rewriting modules that already mitigate such mismatches. Designing a scenario where the text and image are intentionally inconsistent seems artificial and not practically meaningful.
    2. "The condition image can appear at an arbitrary position rather than the first frame of the synthesized video." It is unclear why the authors insist on using first-frame-trained models in this setting. If the desired image position is known (e.g., last or middle frame), existing methods such as Wan, FlexTI2V[1] already support **flexible frame conditioning.** If the position is entirely unknown, the setup itself conflicts with the fundamental principle of conditioned diffusion models, which require **explicit and accurate conditioning.**

2. Beyond the task setup, the method’s main contribution is the MLLM-driven Multi-frame Reasoner, consisting of VisionNarrator and AlignFormer. However, this design is not novel. Prior works such as VideoTetris[2] and VideoDirectorGPT[3] have already used MLLMs to decompose prompts and guide attention across frames. Similar alignment modules are widely adopted in video diffusion models that handle varying prompts. The paper does not clearly explain how its approach differs from these existing methods.

3. The choices of baselines are not convincing. This work is built on Wan 2.1-14B and compared with it. However, for others like LTX-Video and CogVideo, the author selects the smaller versions. The choice of LTX-Video-2B over 13B and CogVideo-2B over 5B is confusing and weakens the experimental comparison.

4. The writing could be substantially improved. There is a lot of redundancy throughout the paper. For example, many explannations of obvious video examples take excessive space. In addition, statements like line 250 — “It is important to note that although the VisionNarrator does not directly participate in the video generation process, its reasoning capabilities are essential for constructing each frame’s context and ensuring the overall coherence of the video.” — are unnecessary. This point is self-evident and describes a very common design.

[1] Lai, Bolin, et al. "Incorporating Flexible Image Conditioning into Text-to-Video Diffusion Models without Training." arXiv preprint arXiv:2505.20629 (2025).
[2] Tian, Ye, et al. "Videotetris: Towards compositional text-to-video generation." Advances in Neural Information Processing Systems 37 (2024): 29489-29513.
[3] Lin, Han, et al. "Videodirectorgpt: Consistent multi-scene video generation via llm-guided planning." arXiv preprint arXiv:2309.15091 (2023).

**Questions:**

1. Why is the auxiliary reconstruction loss not used in the first stage? This ablation appears to be missing.
2. What exactly is Stage 2 of AlignFormer? From the figure, it looks identical to Stage 1 except for another attention layer with the same inputs, and the paper provides no formula or explanation for this stage.
3. Why not compare with video diffusion models that use prompt rewriting or support last-frame generation? The trade-off between retraining a new model and leveraging existing methods should be analyzed.

---

### Official Review · Reviewer_AQbw · 2025-10-29

**Soundness:** 2
**Presentation:** 3
**Contribution:** 3
**Rating:** 6
**Confidence:** 3

**Summary:**

The paper proposes unpaired text–image-to-video generation: the text and the image can describe events at different times, and the image can appear anywhere in the video (not necessarily in the first frame). To handle this, the paper introduces ReasonDiff, which adds two modules on top of a DiT-based video generator. VisionNarrator uses an MLLM to infer a per-frame narrative and the most likely position of the condition image. AlignFormer injects this narrative into the model to predict reasoning-enhanced latents, which are then fused with the condition frame. Experiments on a custom unpaired split of ActivityNet and on MSR-VTT show strong results, especially on text alignment and in a user study.

**Strengths:**

+ Clear motivation: The paper clearly defines the unpaired setting and explains why paired methods fail.

+ Method design: The roles of VisionNarrator and AlignFormer are clear, and the overall pipeline matches the task.

+ Evidence: The quantitative comparisons and ablation studies clearly show the benefits of the method.

**Weaknesses:**

- Quality depends on the MLLM: VisionNarrator drives the core reasoning. Results could be sensitive to the choice of MLLM and prompt design. Please report the prompt variants, ablations with different MLLMs, and failure cases (with examples)?

- Scaling: Training uses 33 input frames, and at inference the model outputs 81 frames. How does the method scale to longer videos or higher FPS? Also, how much latency does VisionNarrator add? Please include a timing and cost breakdown per component.

- Flexibility of the data setup: The unpaired dataset is built by taking the first or last frame as the image and captioning the opposite end as text. How well does ReasonDiff handle multiple events, non-causal relations, or mid-sequence anchors?

- Generalization of the anchor scheme: AlignFormer relies on a single anchor frame. What happpens with multiple condition images or when the image should appear in the middle of the sequence (possibly more than once)?

**Questions:**

See weaknesses.

---

### Official Review · Reviewer_6uFg · 2025-11-02

**Soundness:** 2
**Presentation:** 3
**Contribution:** 2
**Rating:** 4
**Confidence:** 3

**Summary:**

This paper presents ReasonDiff, a reasoning-guided diffusion framework for unpaired text–image to video generation, where the semantic information provided text and image may occur at different timestamps. The approach introduces a VisionNarrator module—a multimodal large language model (MLLM)—that predicts the anchor frame and generates per-frame narrative descriptions. In addition, an AlignFormer with Multi-stage Temporal Anchor Attention (MTAA) integrates these reasoning-aware temporal cues into the diffusion process. Experiments on a custom unpaired benchmark derived from ActivityNet and on MSR-VTT show that ReasonDiff consistently outperforms baselines, including Dynamicrafter, LTX-Video, CogVideoX, and Wan2.1.

**Strengths:**

1. The paper introduces the task of unpaired text–image to video generation, addressing a limitation in existing multimodal systems that assume perfectly aligned inputs.
2. Modular and interpretable architecture. The VisionNarrator and AlignFormer pipeline cleanly separates semantic reasoning from temporal fusion, rather than trying to implicitly teach a diffusion model.
3. Consistent gains in imaging quality, motion smooth, dynamic degree, CLIP-Text, CLIP-Image, and user ranks under the ActivityNet and MSR-VTT. Qualitative examples demonstrate plausible temporal transitions and semantically coherent video generation.
4. The paper systematically studies the contributions of key components—including enhanced latent supervision, multi-prompt narrative guidance, and auxiliary reconstruction losses—providing empirical justification for each design decision.

**Weaknesses:**

1. As the reviewer understands, ReasonDiff is trained on a synthetically unpaired version of the WebVid dataset, while all baseline models (Wan2.1, Dynamicrafter, LTX-Video, CogVideoX) were not trained in this manner. Since these baselines never encountered mismatched modalities, their degraded performance under unpaired evaluation likely reflects domain shift rather than true architectural inferiority. Without retraining baselines such as Wan2.1 on the same unpaired setup, the comparison cannot be considered fair.
2. Although the paper emphasizes that reasoning is central to unpaired text–image-to-video generation, there is no direct assessment of the model’s reasoning ability. The reported metrics cannot verify genuine causal or temporal reasoning. The paper highlights a few successful examples but fails to analyze failure modes. It remains unclear how the method behaves when the MLLM generates physically implausible or logically inconsistent narratives.
3. During training, ReasonDiff uses LLaMA-3.2-11B-Vision-Instruct to generate frame-level narratives, but at inference it switches to gpt-4o for VisionNarrator outputs. No ablation isolates this variable. Moreover, the paper does not justify the switch.
4. The Rewrite Prompt ablation demonstrates that merely expanding or rephrasing the input text using an MLLM conditioned on the image can yield moderate improvements. This overlap implies that ReasonDiff’s gains may largely arise from enhanced textual conditioning rather than from true architectural reasoning mechanisms.
5. The framework divides learning into Stage 1, joint training with the denoising loss, and Stage 2, fine-tuning AlignFormer with an auxiliary latent reconstruction loss. This design appears empirically motivated for stability, but the authors provide no explanation for why end-to-end optimization would fail or how much the two-stage training actually improves performance. Without quantitative comparison to a single-stage baseline, this additional complexity seems weakly justified.

**Questions:**

1. If time permits, could the authors retrain Wan2.1 or another baseline using the same unpaired WebVid setup to ensure fair comparison?
2. How can the model’s reasoning ability be objectively measured? Are there metrics for causal consistency, temporal logic, or narrative coherence that could validate this claim? Can the authors provide examples of where the VisionNarrator MLLM provides a poor, ambiguous, or physically incorrect narrative from the unpaired inputs? How does the ReasonDiff handle such a failure?
3. Why do the authors use GPT-4o for inference instead of the LLaMA-3.2-11B model that was employed during training for narrative generation? Please justify the use of gpt-4o for inference when LLaMA-3.2-11B was used for training data generation. Was it due to LLaMA’s insufficient reasoning quality?
4. What is the actual output of the Rewrite Prompt setting? Are these outputs same as the prompts shown in Appendix Figure 6? If they are not, what would happen if the baseline model (Wan2.1) were given the exact prompts from Appendix Figure 6? If those prompts produce results or performance similar to ReasonDiff, the improvement likely comes from the rewritten textual inputs rather than the proposed reasoning architecture. In that case, the same gains could be achieved simply by a user manually revising or expanding the prompts, which would weaken both the architectural novelty and the underlying motivation for introducing the unpaired reasoning framework.
5. Is two-stage training necessary?

---

### Official Review · Reviewer_g1za · 2025-11-02

**Soundness:** 3
**Presentation:** 3
**Contribution:** 3
**Rating:** 4
**Confidence:** 4

**Summary:**

This paper introduces ReasonDiff, a novel framework for unpaired text-image to video generation, a problem that has received limited attention in the literature. Unlike traditional methods that assume strong alignment between text and image inputs, ReasonDiff addresses the more realistic and challenging scenario where the image and text may describe events that occur at different times or are only loosely related.

**Strengths:**

1. The paper is logically structured and clearly presents its motivation, methodology, and results. The problem setting is novel and practically relevant, as real-world user inputs are often loosely aligned. The use of an MLLM to bridge the semantic gap between unpaired text and image is timely and intuitive, and the integration of reasoning into video generation is well-executed.

2. The experimental results are convincing, showing consistent improvements over baselines like Wan2.1 and CogVideoX. The qualitative examples effectively illustrate the model’s ability to generate coherent and semantically aligned videos.

**Weaknesses:**

1. Lack of innovation: The core idea of inserting a powerful VLM (LLaMA-3.2-11B) to reason over inputs is not architecturally novel. The Text encoder of wan2.1 itself is T5, while the additional VLM inserted in ReasonDiff is LLaMA-3.2-11B. Inserting such a large and strong VLM, it is obvious that this can naturally solve some inherent problems in the original video generation and improve performance.

2. Insufficient Ablations: The paper does not ablate the VLM component. There are no experiments with smaller VLM variants (e.g., 3B or 7B), different architectures, or frozen vs. fine-tuned settings. This makes it unclear whether the full 11B model is necessary or if smaller models could achieve similar performance.

3. Inference Overhead: The use of an 11B-parameter VLM significantly increases inference cost, but the paper does not report latency or memory usage. This is a critical omission for practical deployment and scalability.

4. Limited Baseline Comparison: The paper only compares with video generation models that do not use VLMs (e.g., Wan2.1, CogVideoX). It does not compare with recent VLM-conditioned video generation methods. This limits the ability to assess whether the proposed method is truly superior to other VLM-based approaches.

**Questions:**

Have you experimented with smaller VLM models (e.g., 3B or 7B)? Does performance degrade significantly? If smaller models work nearly as well, this would improve the practicality of your method.

**Details Of Ethics Concerns:**

No, I have no related concerns.

---

### Official Review · Reviewer_XJ9i · 2025-11-02

**Soundness:** 3
**Presentation:** 3
**Contribution:** 2
**Rating:** 4
**Confidence:** 4

**Summary:**

The manuscript proposes an approach for text to video generation where the text may align with any frame in the video (i.e. first frame alignment may not hold true).

Authors broke the tasks into two steps: VisionNarrator and AlignFormer.

**Strengths:**

Authors tackle a new challenge in text to video problem.

Paper is easy to read and follow.

All steps are sound and, IMO, easy to implement.

**Weaknesses:**

I don't believe that the paper brings enough novelty and contribution. There are also some weaknesses in evaluation and data (please see my questions section).

**Questions:**

1- Lines 295-296 and lines 342-343: I believe the authors should test the base generative model on the original datasets and not the specific new version of dataset that they proposed. Audience should know if this whole process is impacting the video generation quality or not. The method may support video generation for a new condition (like unpaired text) but damage the overall quality of the video.

2- Lines 346-348: I believe this will not guarantee unpairing text. For example, many times a description might be valid for all the frames in a video.

3- Lines 311-312: To clarify, is this embedding the visual embedder of standard CLIP model?

---

### Note · Authors · 2025-11-14

I have read and agree with the venue's withdrawal policy on behalf of myself and my co-authors.